# Compact Slot Microring Resonator for Sensitive and Label-Free Optical Sensing

**DOI:** 10.3390/s22176467

**Published:** 2022-08-27

**Authors:** Bingyao Shi, Xiao Chen, Yuanyuan Cai, Shuai Zhang, Tao Wang, Yiquan Wang

**Affiliations:** College of Science, Minzu University of China, Beijing 100081, China

**Keywords:** slot microring resonator, silicon on insulator, biosensors, integrated optics devices

## Abstract

A novel all-pass slot microring resonator (SMRR), intended for label-free optical biosensing based on silicon-on-insulator platforms, is proposed. The sensor consists of a bent asymmetric directional coupler and an asymmetric-slot microring waveguide. The appropriate slot width of 140 nm is identified by the three-dimensional finite-difference time-domain (3D-FDTD) method for better light–matter interaction in applications. According to numerical calculations, the SMRR sensor with a footprint of 10 µm × 10 µm has a concentration sensitivity of 725.71 pm/% for sodium chloride (NaCl) solutions. The corresponding refractive index sensitivity is 403 nm/RIU (refractive index unit), which is approximately six times greater than that of traditional microring resonator sensors. A low detection limit of 0.129% is also achieved. This SMRR is an excellent candidate for label-free optical biosensors due to its compact structure and excellent sensing capability.

## 1. Introduction

Label-free optical biosensors have become increasingly important in environmental monitoring, food safety, medical diagnosis, national defense, and military applications in recent years due to their unique advantages, such as fast response time, immunity to electromagnetic interference, and prospects for mass production [1,2,3,4]. Furthermore, unlike labeled optical biosensors, label-free optical biosensors do not require complicated labeling procedures and long analysis times, making them a viable alternative for simple and cost-effective biosensing applications.

Sensors are made from a variety of materials, including silicon on insulator (SOI), polymers, metals, and chalcogenide glasses, etc. [5]. Now, more interest has been devoted to silicon photonic sensors based on SOI waveguides due to the high refractive index contrast of SOI and its compatibility with complementary metal oxide semiconductor (COMS) technology. It enables the fabrication of cost-effective integrated optics devices. Numerous label-free optical biosensors based on the SOI platforms have been exploited, such as Mach–Zehnder interferometer sensors [6,7], Fabry–Perot resonator sensors [8], surface plasmon sensors [9], grating sensors [10,11], and microring resonator (MRR) sensors [12,13]. To meet the needs of online bio-detection, these sensors are developing toward high integration, high sensitivity, low detection limits, and superior real-time performance. MRRs have attracted extensive research in applications such as optical filters, sensors, lasers, and optical vortex generation due to their compact footprint and high achievable *Q*-factor. The ultra-high *Q*-factor of MRRs has reached up to 10^7^~10^9^, which requires ring waveguides with a bend radius of at least 9.65 mm [14,15,16,17]. A higher *Q*-factor, on the other hand, causes more light energy to be localized in the resonance cavity and weakens the evanescent field outside the waveguide. We need to further reduce the microring’s radius in order to overcome the limitation of a high *Q*-factor. Generally, the bending radius of SOI waveguide-based MRR sensors ranges from a few microns to several hundred microns, which produces evanescent fields that are more sensitive to refractive index changes outside the waveguide.

Sensitivity is one of the most important parameters of sensors. The evanescent field of the resonance mode is used by conventional MRR sensors to identify the sensing medium. The effective refractive index of the resonance mode changes when the sensing medium changes, resulting in detectable changes in the optical properties of the sensing system [18]. Because the resonance mode is predominantly restricted to the silicon layer, typical MRR sensors based on strip waveguides are only sensitive to 70 nm/RIU [19]. In recent years, various structures of MRR sensors have been developed to improve their sensitivity. For example, a one-dimensional photonic crystal MRR with periodic air holes is introduced in a microring waveguide [20,21,22], grating-type MRR [23,24,25], and SMRR [26,27,28,29,30,31,32]. The SMRR introduces an air slot in microring waveguides, where the evanescent field of each individual silicon brick is extremely high, constructively guiding the quasi-TE mode into the slot region to interact with the matter [26]. This unique characteristic makes SMRR sensors more sensitive than traditional MRR sensors.

According to the reference [31], the sensing medium (NaCl solution), however, can only partially fill the slot with a width of 104 nm. The slot widths reported in [26,27,28,29,30] are all smaller than 104 nm, leading to inadequate light–matter interaction in applications. In [33], a biosensor based on concentric microring resonators was proposed with an airgap of 200 nm between the concentric rings. The field distribution is mainly concentrated on the dielectric ring, which is obviously unfavorable for light–matter interaction in the airgap. Therefore, the slot width of the SMRR needs to be further explored. Via the 3D-FDTD method, we designed a novel SMRR sensor with a slot width of 140 nm. A bent asymmetric directional coupler (ADC) was used to efficiently couple the bus waveguide and the asymmetric-slot microring waveguide. Compared to the point-coupled SMRR sensor, the bent ADC offers a better connection [29]. In addition, the bending ADC enables the SMRR to have a more compact structure when compared to SMRR sensors with a straight directional coupler [31]. The sensitivity and limit of detection (LOD) of this device were analyzed for different concentrations of NaCl solutions in detail.

## 2. Structure and Principle

### 2.1. Structure Design

The schematic SMRR sensor is depicted in Figure 1a with a bent-strip bus waveguide acting as input and output ports and an asymmetric-slot microring waveguide as a resonator. It is on an SOI wafer with a 2 μm-thick buried SiO_2_ layer and a 220 nm-thick Si layer. We use a sensing medium (deionized water or other biological buffers) as the top cladding to achieve uniform sensing. Several essential geometrical characteristics are noted on the top view of the SMRR sensor in Figure 1b. To preserve single-mode transmission and better evanescent coupling with the resonator, the bus waveguide width *W*_strip_ is set to 320 nm. Furthermore, the bent component of the bus waveguide has a radius of 5.69 µm, resulting in lower bending loss during the mode transmission [34]. The bending radius of the bus waveguide in coupling region is 5.69 µm, which is denoted as *R*_0_. A bent ADC can be defined as the coupling area where the bus waveguide is closest to the resonator [35]. The gap width between the bus and the resonator is *W*_gap_ and the coupling angle of the bent ADC is *θ*. The bent ADC lengthens the coupling between the bus waveguide and the resonator, allowing for sufficient coupling without increasing the resonance length further. Meanwhile, the *W*_gap_ is as large as 200 nm to avoid fabrication difficulties. The microring radius (*R*), the distance between the center of the rings to the center of the slot microring waveguide, is set to 5 μm. The resonator is made up of the outer ring, air slot, and inner ring; *W*_out_, *W*_slot_, and *W*_in_ represent their widths, respectively. The SMRR proposed in this paper can be manufactured using the fabrication process in [35]. The footprint of our device is 10 μm × 10 μm.

### 2.2. Operation Principle

For an all-pass SMRR, the resonance wavelength supported in a cavity is
(1)$$\lambda_{\mathrm{res}}=\frac{L}{m}n_{\mathrm{eff}}$$
where *L* is the perimeter of the resonance cavity, *m* is the order of the resonance mode (*m* = 1, 2, …), and *n*_eff_ is the effective refractive index of modes. In Equation (1), the change in the refractive index of the cavity surroundings leads to the shift in the resonance wavelength.

The light field transmission inside the SMRR waveguide is shown in Figure 2a. $E_{m}^{+}$ (*m* = 1, 2, 3, 4) represents the incident light amplitude and $E_{n}^{-}$ (*n* = 1, 2, 3, 4) represents the reflected light amplitude in the corresponding position. In Figure 2b, the light field energy is predominantly restricted to the slot area. The transmission of the light field inside the SMRR waveguide can be analyzed using the transfer matrix method (TMM) [36], such that the light field transmission between the bus waveguide and the slot microring waveguide is expressed as
(2)$$\left(\begin{array}{c} E_{2}^{+}\\ E_{4}^{+} \end{array}\right)=\left(\begin{array}{cc} t & jk\;\\ jk & t \end{array}\right)\left(\begin{array}{c} E_{1}^{+}\\ E_{3}^{+} \end{array}\right)$$
where *t* and *k* are the transmission and coupling coefficient, respectively. Because the entire structure is symmetric in the *y*-direction, the light path can be reversed. The relationship between *k* and *t* is characterized as follows, assuming that the loss in the coupling region is ignored.
(3)$$k^{2}{+t}^{2}=1$$

The relationship between the light field amplitude $E_{4}^{+}$ excited in the resonator and the light field amplitude $E_{3}^{+}$ after one cycle of transmission is described as
(4)$$E_{3}^{+}{=e}^{j(\beta +j\alpha )L}E_{4}^{+}{=ae}^{j\eta }E_{4}^{+}$$
where *a* is the linear loss of the resonance cavity, *η* denotes the phase shift of the transmission rate, and *α*, *β* are the transmission loss coefficient and the phase constant, respectively.

Combining Equations (2)–(4), the transmission spectra from the output port without reflection is recorded as
(5)$$\;A={\left|\frac{E_{2}^{+}}{E_{1}^{+}}\right|}^{2}=\frac{a^{2}{+t}^{2}\;-\;2\mathrm{at}\cos\eta }{{1+a}^{2}t^{2}\;-\;2\mathrm{at}\cos\eta }$$

The transmission spectrum of SMRR is plotted in Figure 3 based on Equation (5). For a resonance cavity with zero attenuation (*a* = 1), the transmission of the light that does not satisfy the resonance condition is uniform. Assuming no coupling loss, the transmission of the resonance peak drops to 0. The characteristics of these periodic resonance peaks are identical.

Equation (5) is generally expressed in the logarithmic form.
(6)$${T\;(\lambda )=10\log}_{10}\left(A\right)$$

## 3. Simulation and Analysis

The SMRR sensors were designed using Lumerical’s FDTD solutions, the mode source (fundamental TE mode) was injected into the input port of the bus waveguide, and the transmission spectrum was obtained at the output port. The essential structure parameters, *W*_slot_, *W*_in_, *W*_out_, and *θ*, were optimized sequentially to further improve the sensing performance of the device. It is noted that *Q*-factor and extinction ratio (ER) are two significant indicators of SMRR sensors. *Q*-factor is defined as the ratio of resonance wavelength ($\lambda_{\mathrm{res}}$) to full width at half the maximum of the peak of the resonance ($∆\lambda_{\mathrm{FWHM}}$), that is
(7)$$Q-\mathrm{factor}=\frac{\lambda_{\mathrm{res}}}{∆\lambda_{\mathrm{FWHM}}}$$

The higher the *Q*-factor, the greater the light energy that is localized in the resonance cavity. ER represents the ratio of the maximum to minimum light intensity at the resonance peak shown as follows.
(8)$${\mathrm{ER}=10\log}_{10}\frac{I_{\max}}{I_{\min}}$$

The higher the ER, the more the device is resistant to electromagnetic interference.

The sensitivity and the LOD are the other two indicators of SMRR sensors. The concentration sensitivity ($S_{C}$) is expressed as the wavelength difference change induced by 1 (mass)% concentration (*C*) change:(9)$$S_{C}=\frac{∆\lambda }{∆C}$$
where ∆*λ* is the variation of the resonance wavelength and ∆*C* is the variation of the sensing medium concentration. In addition, the bulk refractive index sensitivity is shown as
(10)$$S_{V}=\frac{∆\lambda }{∆n}$$
where ∆*n* is the refractive index variation of the sensing medium. LOD is defined as the minimum concentration that can be detected, expressed as follows.
(11)$$\mathrm{LOD}=\frac{∆\lambda_{\mathrm{resolution}}}{S_{C}}$$
where $∆\lambda_{\mathrm{resolution}}$ is the resonance wavelength resolution, i.e., 1/15 of the 3 dB bandwidth of the resonance peak [15].

### 3.1. Structure Optimization

The slot microring waveguide confines the light field to the low refractive index region, enhancing light–matter interaction significantly. The slab thickness of the waveguide is set to 220 nm. The sensitivity of the sensor is influenced by *W*_slot_. When *W*_slot_ is small, it is challenging for the sample to enter the slot region, raising the sensor’s *Q*-factor. Meanwhile, the *W*_slot_ should also not be too wide, as this will weaken the electric field intensity in the slot area. Therefore, a straight waveguide with a coupling gap of 100 nm width is used as the bus waveguide to establish the proper *W*_slot_. In Figure 4a, the *Q*-factor and ER are investigated for different *W*_slot_ between 100 and 200 nm in a 10 nm step. It can be seen that the *Q*-factor tends to decrease as the *W*_slot_ increases, while the ER tends to rise. The amount of energy stored in the resonance cavity is represented by the *Q*-factor; however, this does not imply that the sensitivity is proportional to the *Q*-factor. The volume of light–matter interaction also affects the sensor sensitivity. Figure 4b shows that the sensor has a maximum sensitivity of 398 nm/RIU with a *W*_slot_ of 140 nm, and the corresponding *Q*-factor and ER are 869 and 12.7 dB, respectively.

In Figure 5a, *W*_in_ and *W*_out_ are adjusted to 260 nm. Due to the bending effect of the slot microring waveguide, the center of the electric field is biased towards the outer ring, which causes a larger bending loss and weakens the interaction of the light with the external sensing medium in the slot region. We employ the asymmetric-slot microring waveguide structure to reduce the bending impact [5,30]. By designing *W*_in_ larger than *W*_out_, it is feasible to propagate the resonant light along the slot center and raise the light intensity in the slot region [37,38]. According to Figure 5b, when *W*_in_ and *W*_out_ are set to 270 nm and 250 nm, respectively, the electric field energy is mostly concentrated in the slot region, and the electric field distribution in the inner and outer rings is more uniform, reducing the bending loss. *W*_in_ and *W*_out_ are therefore adjusted to 270 nm and 250 nm, respectively.

The bent ADC is optimized to maximize the evanescent coupling between the bus waveguide and the slot microring waveguide [35,39]. The coupling length is determined by *θ*. The bent ADC lengthens the coupling region in comparison to conventional point coupling, and can achieve critical coupling with a wider coupling distance *W*_gap_. Hence, the *W*_gap_ between the bent bus waveguide and the slot microring is expanded to 200 nm to satisfy the processing precision requirements. The dependence of *Q*-factor and ER on the coupling angle *θ* is plotted in Figure 6. The *Q*-factor is inverse to *θ* due to the higher bending loss of the bus waveguide. On the other hand, ER tends to increase with larger *θ*, demonstrating an increase in the coupling effect. In order to balance the two, *θ* is determined to 36°, thus the *Q*-factor and ER are 1207 and 14.18 dB, respectively. The optimized geometrical parameters for the SMRR are listed in Table 1.

### 3.2. Performance Analysis

After obtaining the optimized geometric parameters of SMRR, the transmission spectrum at the output port in a deionized water (refractive index *n*= 1.333) environment is shown in Figure 7a. Compared to the transmission spectrum derived from Equation (5), the transmission spectrum obtained from the simulation has a transmission rate of less than 0 dB (corresponding to the normalized transmission of 1) for the non-resonance peaks; there is propagation loss in the transmission of light in the waveguide. The minimum transmission of the resonance peak is a definite value, indicating the presence of coupling loss here. The propagation loss of slot waveguides with water as the top cladding was estimated to be 21.5 dB/cm in [31]. The bending loss of slot waveguides with a bending radius of 5 µm was 160 dB/cm. Reducing losses is a crucial issue for future work. We also replaced the original bus waveguide with a 320 nm-width straight waveguide; its transmission spectrum is shown in Figure 7b. By comparing the transmission spectra of the two, the resonance peak of the transmission spectrum of the SMRR with a bent ADC proposed in this paper has a greater ER. This indicates that the bent ADC has lower coupling losses.

The resonance peaks are labeled as modes A–D from short to long wavelengths. The *Q*-factors for modes A–D are 1624, 1207, 1113, and 903, correspondingly. Resonance peaks shift when different liquids are applied to the upper surface of SMRR. It is reported that the refractive index of an aqueous solution of NaCl varies by 0.0018 RIU per mass % at a temperature of 20 °C [19]. Here, the concentrations of NaCl solutions used in simulations are 1%, 2%, 3%, 4%, and 5%, which correspond to refractive indexes of 1.3348, 1.3366, 1.3384, 1.3402, and 1.342, respectively. This is realized by changing the background refractive index to switch the concentration of NaCl solutions. Figure 8a shows the dependence of resonance wavelengths A-D on the concentration of the NaCl solutions. The concentration sensitivity *S*_C_, obtained in four resonance modes, is 662.57 pm/%, 696.86 pm/%, 725.71 pm/%, and 757.43 pm/%, while the index sensitivity *S_V_* is 368 nm/RIU, 387 nm/RIU, 403 nm/RIU, and 421 nm/RIU, respectively. If mode C is picked as the sensing peak, the LOD of the SMRR sensor is 0.129% at 1564 nm. For optical sensing, the minimum refractive index is also one of important specification [31]. The detectable 0.129% concentration change corresponds to a 2.32 × 10^−4^ RIU refractive index change. The shift in resonance peak at 1564 nm as a function of concentrations of NaCl solutions is shown in Figure 8b. The electric filed profile is depicted in Figure 9, with most of the energy being transmitted in the slot region.

Table 2 shows an extended comparison of several configurations based on MRR. Label-free optical sensors based on MRR have been currently employed to detect various substances such as proteins, DNA, NaCl solutions etc. The *S*_V_ value for these sensors is approximately 10^2^. MRR sensors make on-chip sensing easy to integrate due to its small footprint. High integration is an advantage of SMRR sensors based on SOI waveguides over GeSbSe SMRR sensors. For SOI-based SMRR sensors, the slot width in ring waveguides is typically below 104 nm, which makes it easy to achieve higher Q-factors. However, it is unfavorable for external substances to inject into the slot region, which causes applications to be less sensitive. As the slot width is 200 nm or greater, the light field is dispersed in the silicon ring, weakening the light–matter interaction. As a result, a 140 nm-width slot in our proposed SMRR is better suited to facilitate light–matter interactions, improving sensing performance. Furthermore, the sensor’s performance is typically inversely correlated with its footprint, so the high sensitivity is obtained at the expense of the device miniaturization [32]. To further improve the light–matter interaction without increasing the device’s footprint, we bend the ADC structure and enlarge the slot of the SMRR sensor to 140 nm. The suggested SMRR sensor’s sensitivity has been increased to 403 nm/RIU, offering considerable potential for label-free optical biosensing applications.

## 4. Conclusions

In summary, we present a compact SMRR based on the SOI platform for label-free optical biosensing. The structural parameters of the asymmetric slot microring waveguide and bent ADC are designed to obtain a high sensing performance. The slot with a width of 140 nm is more effective for light–matter interaction than other SMRR sensors, and bent ADC enhances the coupling between the bus waveguide and the slot microring waveguide, while maintaining the miniaturization of the device. The results show that the sensor has a concentration sensitivity of 725.71 pm/% and a refractive index sensitivity of 403 nm/RIU, which is nearly six times higher compared to traditional MRR sensors. The sensor has a low LOD of 0.129%. In addition, we will further apply other chemical liquids and biological samples to investigate the optical sensing performance of the SMRR sensor by experiment. The SMRR sensor will be essential in the field of label-free optical biosensing because of its small sensor footprint (10µm × 10µm), which makes it easier to integrate with other optical components on SOI chips.

## Figures and Tables

**Figure 1 sensors-22-06467-f001:**
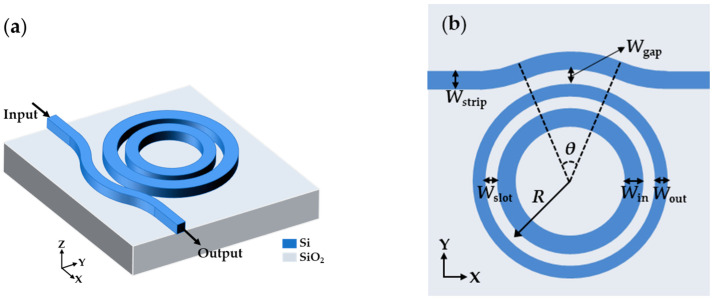
(**a**) Schematic of SMRR sensor. (**b**) Top view of the device.

**Figure 2 sensors-22-06467-f002:**
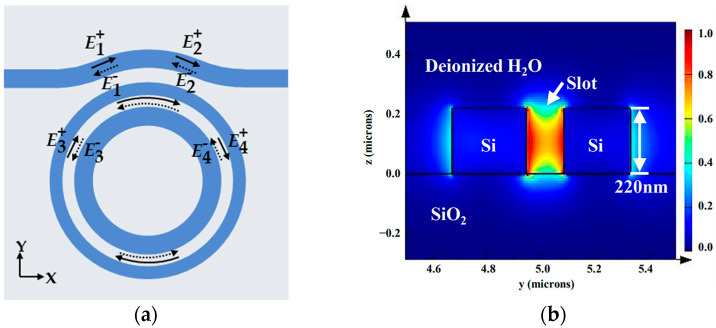
(**a**) Light field propagation inside the SMRR sensor. (**b**) Mode profile for fundamental quasi-TE mode inside slot microring waveguide in deionized water environment.

**Figure 3 sensors-22-06467-f003:**
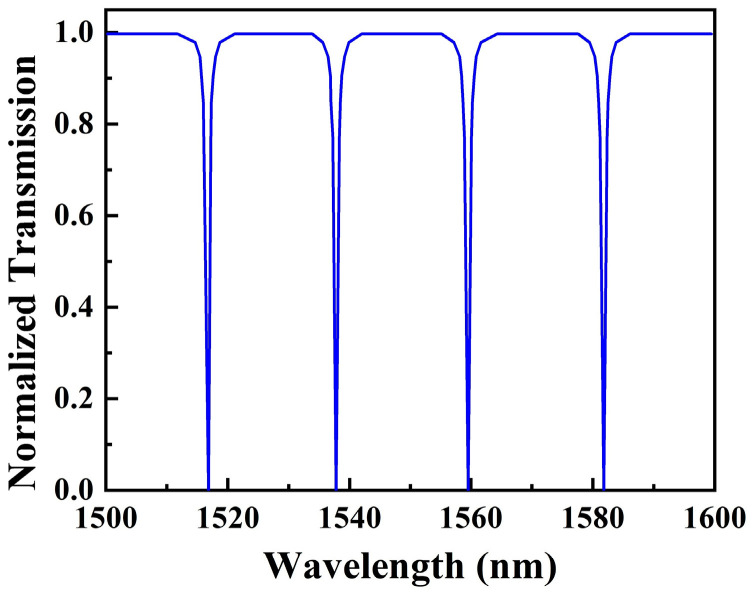
Transmission spectrum of the proposed SMRR with *a* = 1.

**Figure 4 sensors-22-06467-f004:**
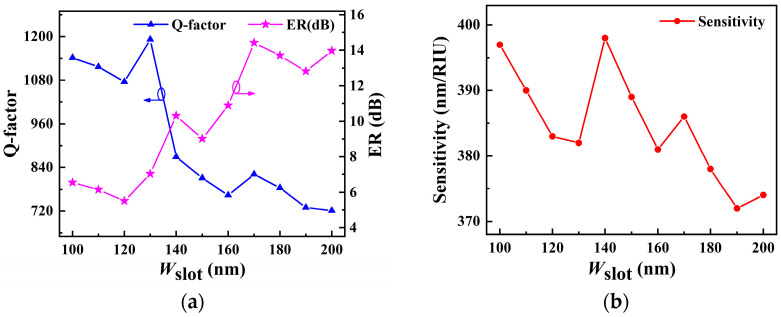
(**a**) Dependence of *Q*-factor and ER of SMRR on *W*_slot_ at resonance wavelength. (**b**) Sensitivity as a function of *W*_slot_ at the resonance wavelength.

**Figure 5 sensors-22-06467-f005:**
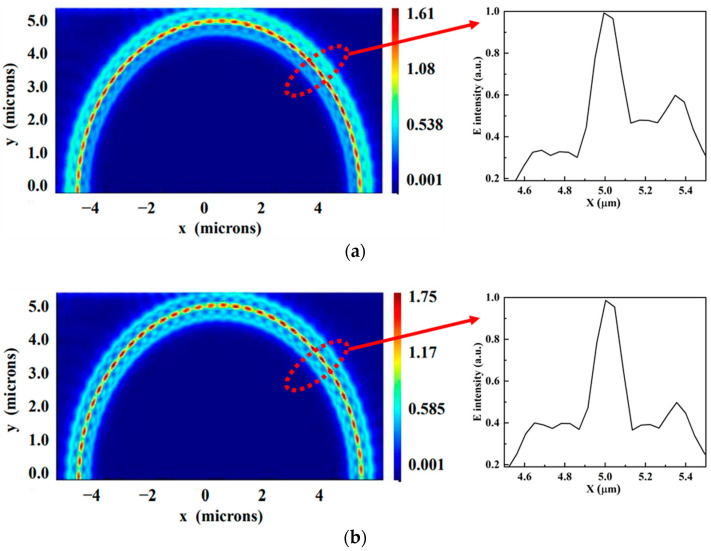
(**a**) Electric field distribution with *W*_in_ = *W*_out_ = 260 nm. (**b**) Electric field distribution with *W*_in_ = 270 nm, *W*_out_ = 250 nm. The inserts show the normalized electric field distribution of the X cross section.

**Figure 6 sensors-22-06467-f006:**
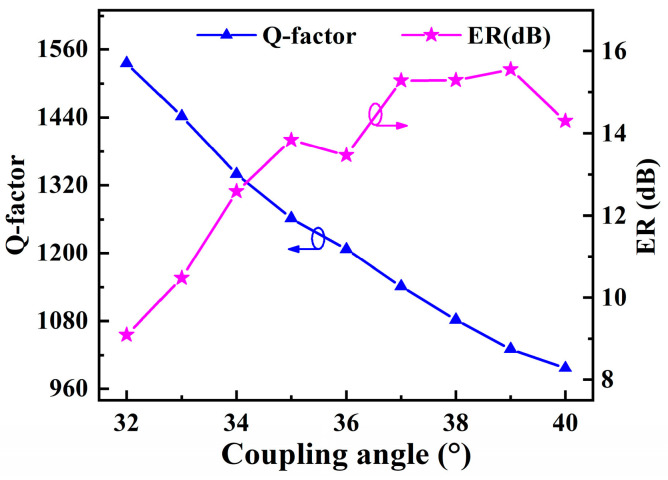
Dependence of *Q*-factor and ER of SMRR on coupling angle at the resonance wavelength.

**Figure 7 sensors-22-06467-f007:**
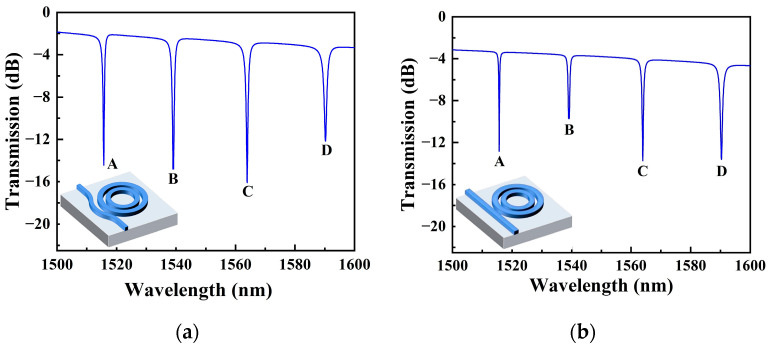
The transmission spectra of SMRR. (**a**) Bent and (**b**) straight waveguides are used as bus waveguides, respectively.

**Figure 8 sensors-22-06467-f008:**
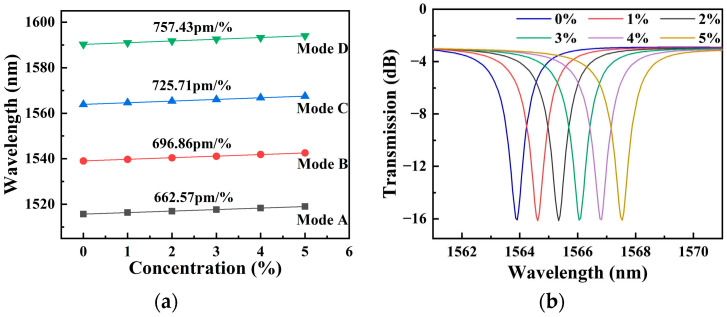
(**a**) Shift of the resonance wavelength with different concentrations of NaCl solutions. (**b**) Shift in resonance peak at 1564 nm with different concentrations of NaCl solutions.

**Figure 9 sensors-22-06467-f009:**
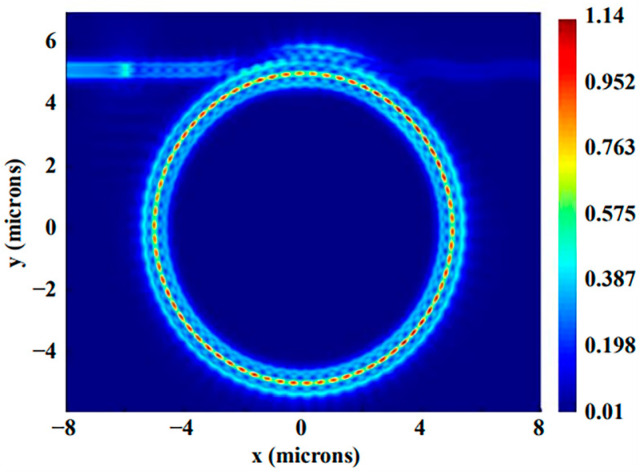
Electric field profile of mode C.

**Table 1 sensors-22-06467-t001:** Geometrical parameters of SMRR.

Parameter	Value
Bus waveguide width (*W*_strip_)	320 nm
Radius of bent-strip waveguide (*R*_0_)	5.69 μm
Outer ring waveguide width (*W*_out_)	250 nm
Slot width (*W*_slot_)	140 nm
Inner ring waveguide width (*W*_in_)	270 nm
Ring radius (*R*)	5 μm
Waveguide height (*h*)	220 nm
Coupling gap (*W*_gap_)	200 nm
Coupling angle (*θ*)	36°

**Table 2 sensors-22-06467-t002:** Comparison of different MRR sensors.

Structure	Footprint	*W*_slot_ (nm)	*Q*-Factor	Analyte Detected	*S*_V_ (nm/RIU)	LOD	Reference	Note
GeSbSe SMRR	120 μm × 120 μm	50	10,000	NaCl solutions	471	3.3 × 10^−4^ RIU	[5]	
SOI MRR	13 μm × 10 μm	\	20,000	Proteins	70	10 ng/ml	[19]	
Photonic crystal MRR	\	\	9300	\	200	\	[22]	Simulation
Grating-type MRR	\	\	12,900	Glucose solutions	363	5.46 × 10^−5^ RIU	[25]	
Double-slot-waveguide-based MRR	25 μm × 15 μm	100	580	NaCl solutions	708	\	[27]	Simulation
SOI SMRR	15 μm × 8.5 μm	100	2000	NaCl solutions	297.13	1.1 × 10^−4^ RIU	[28]	
SOI SMRR	12 μm × 12 μm	60	126,133	\	480.4	2.6 × 10^−5^ RIU	[29]	Simulation
SOI SMRR	\	100	\	Glucose solutions	360	\	[26]	Simulation
SOI SMRR	13 μm × 10 μm	104	330	Proteins	298	4.2 × 10^−5^ RIU	[31]	
Concentric dual-MRRs	27.646 μm^2^	200 (ring-ring air gap)	\	DNA	683	\	[33]	Simulation
	\	400 (ring-ring air gap)	\	Sucrose solutions	180	1.1 × 10^−5^ RIU	[40]	Simulation
SOI SMRR	60 μm × 60 μm	120	1900	Chemical liquids	476	1.05 × 10^−5^ RIU	[32]	
SOI SMRR	10 μm × 10 μm	140	1113	NaCl solutions	403	0.129%	Our work	Simulation

## Data Availability

Not applicable.

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
