# Peer review of "Compact Slot Microring Resonator for Sensitive and Label-Free Optical Sensing"

_sensors, 2022, doi:10.3390/s22176467_

Round 1

Reviewer 1 Report

Shi et al. have designed a (microring resonator)-based optical sensor, and have tested its performance with model NaCl solutions. The principle of operation, and the key performance parameters of (microring resonator)-based sensors are described in detail, and this makes the manuscript interesting and useful for the readers. The manuscript, however, requires revision. The following comments should be addressed.

1. The authors claim that the sensor developed is designed for biosensor applications. In the experiments reported, only NaCl solutions, devoid of any biological molecules or particles (such as proteins, nucleic acids, or viral particles), have been tested. In this connection, the following specifications should be made:

- the manuscript title should be changed. "Biosensing" should not be mentioned in the title. For instance: "Compact Slot Microring Resonator for Sensitive and Label-free Optical Sensing";

- in the abstract, the first sentence should be specified in order to emphasize its intended application (though no experiments with biological objects have been performed), and split into two sentences to make it better understandable: "A novel all-pass slot microring resonator (SMRR), intended for label-free optical biosensing based on silicon-on-insulator platform, is proposed. The sensor consists of a bent asymmetric directional coupler and an asymmetric slot microring waveguide."

- in the conclusions, the authors should emphasize that further experiments are required in order to investigate the performance of the sensor upon biosensing.

2. Careful editing of English language is recommended, and grammatical errors should be corrected throughout the text. For instance:

- L. 14-15: "...and refractive index sensitivity of 403 nm/RIU, which is roughly 6 times that conventional microring resonator sensors." Something is wrong with this phrase. I recommend to check this phrase, and to split the whole sentence (L. 12-15) into several separate sentences, as currently it is unnecessarily long and unclear.

- L. 181-184, expected: "The concentration sensitivity SC, obtained in four resonance modes, was 662.57 pm/%, 696.86 pm/%, 725.71 pm/%, and 757.43 pm/%, while the index sensitivity SV was 368 nm/RIU, 387 nm/RIU, 403 nm/RIU, and 421 nm/RIU, respectively."

3. In analytical chemistry, the lower limit of detection (LOD) is commonly expressed in mass (ng, pg, etc.) or concentration (mg/L, M, etc., and the concentration units are commonly accepted in biosensor analysis) units, but not "as the minimum refractive index that can be detected" (as stated by the authors on P. 4, L. 124-125). Moreover, the Eq. 11 seems to be misleading, since the subscript index of S is missing: do the authors mention Sc or Sv here? According to the authors' statement on L. 124-125, Sv is expected. However, since in biosensing, LOD is commonly expected in concentration units, Sc is expected, so that LOD will be in the units of  C.

4. In the Abstract (on L. 13), and in the Conclusions (on L. 211), the authors should specify whether mass % of NaCl is mentioned as the concentration unit.

Sincerely,

The reviewer

Reviewer 2 Report

The Authors propose an all-pass slot microring resonator for label-free optical biosensing based on SOI platform. The performance seems interesting in comparison with other ring resonators, but an overview of integrated biosensors should be enlarged. According to the following comments, I could provide a rating after the Authors’ responses to the following comments:

-          For ring resonators, the Q-factor plays a crucial role to maximize the sensor sensitivity. Therefore, in the first section, high Q factor ring resonators should be reported (e.g. Integrated waveguide coupled Si 3 N 4 resonators in the ultrahigh-Q regime. Optica, 1(3), 153-157,  2014; Rigorous model for the design of ultra-high Q-factor resonant cavities. IEEE, 2016; Ultralow 0.034 dB/m loss wafer-scale integrated photonics realizing 720 million Q and 380 μW threshold Brillouin lasing. Optics letters, 47(7), 1855-1858, 2022). To overcome the Q-factor limitation, the Authors should stress on the benefits of the evanescent field, by comparing the performance of the proposed sensor (based on slot waveguide) with a MRR sensor with same footprint. Simulations of comparison could be useful.

-          The novelty of the proposed device should be stressed. In particular, the literature is full of slot ring resonator, and the tapered coupling section has been proposed in Integrated waveguide coupled Si 3 N 4 resonators in the ultrahigh-Q regime. Optica, 1(3), 153-157,  2014. About that, all parameters of this region should be reported. Moreover, the mathematical model used to describe the e.m. response of the proposed device is similar to New microwave photonic filter based on a ring resonator including a photonic crystal structure. In 2017 19th International Conference on Transparent Optical Networks (ICTON) (pp. 1-4), 2017. IEEE. However, the presence of the terms Ei(-) in Figure 2 is not clear.

-          An estimation of the losses taken into account in the simulations is not provided. They strongly affect the device performance. Also the losses of the surrounding medium have been neglected. Please consider all types of losses (medium and device), also including propagation (of the order of dB/cm) and curvature losses.

-          As previously reported, the performance of the proposed device should be compared with other configurations one (e.g., see papers of Letchuga’s or Krauss’s groups). An extended comparison table could help the reader to rate the reported performance.

Reviewer 3 Report

The authors proposed a micro ring resonator for optical biosensing application. Although this manuscript is interesting, there are many problems, such as English problems, that need to be carefully addressed:

 ¾   The authors should indicate what software package they are using for the FDTD simulations. If it was developed by them, there should be a reference to a publication.

¾   The authors need to add a transmission diagram of Fig. 1. (Transmission vs. wavelength)

¾   In Fig. 6, why peak D has a better sensitivity value?

¾   The English is poor in this manuscript; there are some English problems.

¾   In the Abstract, it is not clear what the reported abbreviation RIU means.  Later in the text becomes clear that this abbreviation.

¾   In the Abstract, it is not clear what the reported abbreviation FDTD means.

¾   The fabrication process of the structure should explain by the authors.

¾   The authors have not discussed the details of the simulation settings.

¾   The geometrical values of all parameters in Fig. 1 should be given in the table.

¾   The authors need to compare their simulation results with the theory stated in section 2.2.

¾   The authors need to compare their transmission in simulation results with Eq. (5).

¾   The authors need to add a transmission graph of the structure without bending the input waveguide and compare it with the proposed structure.

¾   The following paper is a very similar structure to the proposed manuscript. Why haven't the authors referenced this article?

o   Li, Xiaohui, et al. "Sensitive label-free and compact biosensor based on concentric silicon-on-insulator microring resonators." Applied optics 48.25 (2009): F90-F94.

Round 2

Reviewer 2 Report

The Authors have modified the manuscript according to the Reviewer comments. I suggest the publication.

Reviewer 3 Report

accept

This manuscript is a resubmission of an earlier submission. The following is a list of the peer review reports and author responses from that submission.